Validation and analysis of the geographical origin of Angelica sinensis (Oliv.) Diels using multi-element and stable isotopes

Li Shanjia lishanjia@lut.edu.cn 1 2
Wang Hui 1
Jin Ling 3
White James F. 4
Kingsley Kathryn L. 4
Gou Wei 1
Cui Lijuan 1
Wang Fuxiang 1
Wang Zihao 1
Wu Guoqiang 1
1 School of Life Science and Engineering, Lanzhou University of Technology , Lanzhou , Gansu , China
2 Key Laboratory of Land Surface Process and Climate Change in Cold and Arid Regions, Northwest Institute of Eco-Environment and Resources, Chinese Academy of Sciences , Lanzhou , Gansu , China
3 College of Pharmacy, Gansu University of Chinese Medicine , Lanzhou , Gansu , China
4 Department of Plant Biology, Rutgers University , New Brunswick , United States of America
Nelson David
Electronic publication date: 2021 Aug 6
Publication date: 2021
Volume: 9
Electronic Location ID: e11928
Received 2020 Dec 7; Accepted 2021 Jul 17
Copyright: ©2021 Li et al.
Copyright year: 2021
Copyright holder: Li et al.
License: This is an open access article distributed under the terms of the Creative Commons Attribution License, which permits unrestricted use, distribution, reproduction and adaptation in any medium and for any purpose provided that it is properly attributed. For attribution, the original author(s), title, publication source (PeerJ) and either DOI or URL of the article must be cited.
License URL: https://creativecommons.org/licenses/by/4.0/

Keywords: Angelica sinensis, Mineral elements, Stable isotopes, Discriminant analysis

Funding: The National Natural Science Foundation of China No. 41961007 The Gansu Provincial Key Research and Development Program No. 18YF1FA066 The Lanzhou Science and Technology Development Program No. 2017-4-94 This work was supported by the National Natural Science Foundation of China (No. 41961007), the Gansu Provincial Key Research and Development Program (No. 18YF1FA066), and the Lanzhou Science and Technology Development Program (No. 2017-4-94). The funders had no role in study design, data collection and analysis, decision to publish, or preparation of the manuscript.

==============================
Background

Place of origin is an important factor when determining the quality and authenticity of Angelica sinensis for medicinal use. It is important to trace the origin and confirm the regional characteristics of medicinal products for sustainable industrial development. Effectively tracing and confirming the material’s origin may be accomplished by detecting stable isotopes and mineral elements.

Methods

We studied 25 A. sinensis samples collected from three main producing areas (Linxia, Gannan, and Dingxi) in southeastern Gansu Province, China, to better identify its origin. We used inductively coupled plasma mass spectrometry (ICP-MS) and stable isotope ratio mass spectrometry (IRMS) to determine eight mineral elements (K, Mg, Ca, Zn, Cu, Mn, Cr, Al) and three stable isotopes (δ13C, δ15N, δ18O). Principal component analysis (PCA), partial least square discriminant analysis (PLS-DA) and linear discriminant analysis (LDA) were used to verify the validity of its geographical origin.

Results

K, Ca/Al, δ13C, δ15N and δ18O are important elements to distinguish A. sinensis sampled from Linxia, Gannan and Dingxi. We used an unsupervised PCA model to determine the dimensionality reduction of mineral elements and stable isotopes, which could distinguish the A. sinensis from Linxia. However, it could not easily distinguish A. sinensis sampled from Gannan and Dingxi. The supervised PLS-DA and LDA models could effectively distinguish samples taken from all three regions and perform cross-validation. The cross-validation accuracy of PLS-DA using mineral elements and stable isotopes was 84%, which was higher than LDA using mineral elements and stable isotopes.

Conclusions

The PLS-DA and LDA models provide a theoretical basis for tracing the origin of A. sinensis in three regions (Linxia, Gannan and Dingxi). This is significant for protecting consumers’ health, rights and interests.

Introduction

Angelica sinensis is a native Chinese plant (Ross, 2001). Its dried root, commonly known as Danggui in China, is frequently used in Chinese traditional medicines. Its use was first recorded in the Divine Farmer’s Classic of Materia Medica (also known as Shennong Bencao Jing) more than two thousand years ago (Ai et al., 2013). It is widely distributed in China, Korea, Japan, Europe and America (Lu et al., 2020; Mei et al., 2015; Zhao et al., 2003). The Dingxi region in the Gansu Province of China is the primary A. sinensis production area, accounting for 70% of China’s total production and 80% of its export (Giacomelli et al., 2017). A. sinensis is often used to treat gynecological diseases (Circosta et al., 2006) and is also called “female ginseng” (Hook Ingrid, 2014; Tian et al., 2017). A. sinensis is becoming increasingly popular worldwide as a health supplement for women.

Recently, food and drug safety issues have drawn much attention due to increased global import and export trade. However, we lack effective scientific tools and techniques to ensure the enforcement of laws and regulations (Tang et al., 2015). The geographical origin of A. sinensis has been unexplored so far. Previous research has mainly focused on the extraction and separation of natural compounds from A. sinensis and screening their bioactivity (Chao & Lin, 2011; Jin et al., 2012). More than 70 compounds have been isolated from the dried roots of A. sinensis (Chao & Lin, 2011), of which ferulic acid and Z-lignoside are the two main identified components (Giacomelli et al., 2017; Lu et al., 2005). The molecular genetics of A.sinensis are not well studied (Mei et al., 2015; Zhao et al., 2003). Molecular genetics can explain the genetic distance between A. sinensis varieties, but it cannot be used as evidence of geographic distance (Adamo et al., 2012). Thus our ability to trace its geographic origin is limited.

Many countries have issued laws and regulations for food and drug traceability, resulting in the development of radio frequency identification (RFID), barcode readers (George et al., 2019) and two-dimensional barcode technology (Chen et al., 2020). Although these technologies can track food source and flow, it is only limited to a document label description. It is impossible to guarantee whether the label matches the product in complex product transportation and processing chains. Furthermore, the provenance of food traceability must also be verified. The geographical tracing of food and drugs typically occurs through mineral elements (Coelho et al., 2019; Sayago, González-Domínguez & Beltrán, 2018; Potortì et al., 2018), stable isotopes (Camin et al., 2017; Pianezze et al., 2019; Wadood, Guo & Wei, 2019; Zhou et al., 2019), organic compounds (Lukić et al., 2018; Fang et al., 2019; Liu et al., 2017), and infrared spectral absorption characteristic peaks (Wang et al., 2019; Hu et al., 2019; Innamorato et al., 2019). Tracing based on mineral elements and stable isotopes has been used for a variety of plant and animal products with geographical origin traceability. For example, tea (Deng et al., 2019; Zhang et al., 2018), pear (Albergamo et al., 2018), rice (Liu et al., 2018), olive oil (Damak et al., 2019), alcohol (Geană et al., 2017; Pepi et al., 2019; Wu et al., 2019), meat (Hao et al., 2019) and aquatic products (Han et al., 2020; Luo et al., 2019; Li et al., 2018) have been traced using these methods.

Mineral elements in soil are significantly related to corresponding levels in plants, in respect of the traceability of agricultural raw materials (Greenough, Mallory-Greenough & Fryer, 2005; Brzezicha-Cirocka, Grembecka & Szefer, 2016; Catarino et al., 2008). The mineral elements in A. sinensis reflect its contact with soil geochemistry. The regional geological heterogeneity could lead to variations in the mineral-element contents of A. sinensis. Mineral elements are mostly stable in the raw materials of agricultural products compared with other components, which makes this a promising identifier for determining food provenance (Zhao, Zhang & Zhang, 2017; Catarino et al., 2018). Stable isotopes of plants are affected by environmental factors and its temporal and spatial specificity. It has been reported that δ13C and δ18O are closely related to factors such as precipitation, temperature, altitude, and slope (Hultine & Marshall, 2000; Granath et al., 2018). Among them, δ13C values depend on the relative CO2 value in the external environment and the CO2 concentration of the intercellular environment (mainly stomatal density and stomatal conductance) (Ferrio, Voltas & Araus, 2003). δ18O values in plants mainly depend on atmospheric precipitation and transpiration (Ripullone et al., 2008). Precipitation and radiation show temporal and spatial specificity, which leads to differences in the δ18O enrichment in plants. δ15N values in plants are mainly affected by nitrogen pools in soil. δ15N typically exists in the form of nitrogen compounds. The nitrogen compounds from different soil types lead to variations of δ15N across different geographical regions through nitrification, denitrification and ammonia volatilization (Handley & Raven, 2010). Therefore, the spatial distribution of stable isotope ratios in the environment is necessary to determine the A. sinensis’ isoscapes. This study provides a theoretical basis for evaluating the geographical origin of A. sinensis based on the calibrated characteristic values of a stable isotope.

In this study, 25 samples of A. sinensis were collected from three regions (Linxia, Gannan, and Dingxi) in the southeastern province of Gansu, China. We measured three stable isotopes and eight mineral elements. Principal component analysis (PCA), partial least square discriminant analysis (PLS-DA) and linear discriminant analysis (LDA) for multivariate data discriminant analysis were conducted. We sought to (1) describe the differences between stable isotopes and mineral elements in three different regions (Linxia, Gannan and Dingxi); (2) screen the landmark mineral elements and stable isotope factors of A. sinensis in the different regions; and (3) compare PCA, PLS-DA and LDA’s discrimination and cross-validation ability on A. sinensis.

Materials & Methods

Study area and sample collection

Samples were taken from the Linxia (LX), Gannan (GN) and Dingxi (DX) regions of southeast Gansu Province, China, between 34°24′–35°57′N and 103°11′–104°28′E. Linxia has a semi-arid climate, with an average annual temperature of 5.2–7.0 °C and an average annual precipitation of 350–660 mm. Gannan has a plateau climate, with large regional differences in annual average temperature and extremely uneven geographical distribution of precipitation; the average annual temperature is between 1–13 °C and the average annual precipitation is 518–634 mm. Dingxi is a middle temperate semi-arid area with a continental monsoon climate, with an average annual temperature of 5.5 °C and an average annual precipitation of 635 mm.

We collected 25 samples from farmlands during the A. sinensis harvest season from April to May 2019. We contacted local farmers to obtain their permission to collect experimental samples. Global Positioning System (GPS) was used to record the longitude and latitude of each sampling plot, and ArcGIS (Version 10.7) was used to plot the sampling points. As shown in Fig. 1, samples were collected from planting sites (including five in Linxia, seven in Gannan, and 13 in Dingxi). We dug the roots of A. sinensis, gently brushed away the surface soil, and stored the sample in a self-sealing bag. Samples were put into a 4 °C fresh-keeping container for refrigeration. A freshly collected sample was cleaned and rinsed with deionized water three times then dried in a constant temperature oven at 70 °C to a constant weight. Dried roots were crushed by a high-speed pulverizer and were passed through a 100-mesh sieve. These samples were placed in self-sealing bags for storage.

Figure 1 Geographical distribution of A. sinensis sampling areas in three regions (Linxia, Gannan and Dingxi).

Stable isotope measurement

We weighed 5.0 mg of ground A. sinensis into tin capsules. We used an isotope ratio mass spectrometer (IRMS Delta plus XP; Thermo-Fisher, USA) and an element analyzer (Flash EA) to analyze δ13C and δ15N. The sample was introduced into the element analyzer through the autosampler and was transferred to the IRMS through the carrier gas (helium) to determine δ13C and δ15N. Reference materials, including USGS24, IAEA-600, IAEA-N-2 and IAEA-NO-3, were used for calibration. During the analysis, a laboratory standard was interspersed for every 12 samples for calibration. The long-term accuracy of the instrument was 0.2‰. We used a known laboratory wheat flour (B2159) standard with δ13C (δ13CV−PDB =−13.68 ± 0.2 (‰)) and δ15N (δ15NAir =1.58 ± 0.15 (‰)) values to check the accuracy of the instrumental condition. The samples were analyzed in triplicate. We calculated the average of the three samples.

In order to perform δ18O analysis, we weighed 1 mg of the dried A. sinensis sample and packed it into an isotope silver capsule. Samples were loaded into the elemental analyzer Flash EA through the autosampler, and the analyte was sent to the isotope ratio mass spectrometer (253plus; Thermo-Fisher, USA) for δ18O determination. The international reference standards for δ18O are IAEA-601and USGS55. The samples were analyzed in triplicate. We calculated the average of the three samples.

The following formula was used to calculate the relative deviation of the measured δ13C, δ15N and δ18O from the ratio of international standard substances: δ‰=Rx−RstdRstd×1000

—δ(‰): The heavier isotope values in the sample (13C, 15N, 18O);

Rx: The ratio of stable isotope in the sample (13C/12C, 15N/14N, 18O/16O);

Rstd: International standard material stable isotope ratio ((13C/12C)VPDB, (15N/14N)Air, (18O/16O)VSMOW)

Mineral elements analysis

All of the materials used for standard solution and sample processing and the Teflon digestion vessel were immersed in 20% HNO3 solution for 24 h, then rinsed with deionized water and dried before testing the standard solution and samples. We used inductively coupled plasma mass spectrometry (ICP-MS) tuning solution to calibrate ICP-MS. The Rh standard solution was used as an internal standard. We prepared the working standard solution by diluting the stock standard solution of K, Mg, Ca, Zn, Cu, Mn, Cr and Al with 2% HNO3. The preparation concentration of K, Mg and Ca was 0–150 mg/L, and the preparation concentration of Zn, Cu, Mn, Cr and Al was 0–500 µg/L. Each working standard solution tested was repeated three times to obtain a standard curve. The standard curve R2 of all the elements measured was greater than 0.9990.

We weighed 0.3 ± 0.01 g of the A. sinensis powder. These samples were then placed into a microwave digestion tank and 8 mL of concentrated HNO3 was added. The container was covered and digested. After digestion was complete, the A. sinensis sample was cooled to room temperature in a 25 mL volumetric flask to a constant volume, and this was repeated three times. The digestion process was: (1) 800 W constant microwave power at 100 °C for 10 min; (2) 800 W constant microwave power at 150 °C for 10 min; (3) 800 W constant microwave power at 180 °C for 30 min.

ICP-MS was used to determine eight mineral elements in A. sinensis samples (Albergamo et al., 2018) and their quantification limits were 0.003 mg/L (K), 0.008 µg/L (Ca), 0.0008 mg/L (Mg), 2.830 µg/L (Al), 0.062 µg/L (Zn), 0.023 µg/L (Mn), 0.443 µg/L (Cu), and 0.026 µg/L (Cr). The limit of quantification (LOQ) was the analyte concentration equivalent to 10 times the standard deviation (10 σ) of the blank solution for 10 consecutive measurements.

Statistical analysis

We analyzed the significant differences between variables form the three different regions after testing the 11 variables’ distribution and the analysis variables’ basic statistics. These were statistically analyzed by grouping (P < 0.05), and multiple comparisons between paired groups were performed by using Tukey’s honestly significant differences (Tukey HSD) method. We standardized the data and performed dimensionality reduction and discriminant analysis.

The traceability and provenance of food depends on reliable measurement data and using the appropriate chemometric data processing methods (Bertacchini et al., 2013). Single element stoichiometry is greatly limited in terms of traceability, and it is impossible to comprehensively evaluate a large number of variables with it. Thus, a lot of important traceability information is lost. Multivariate data analysis techniques such as PCA, PLS-DA, and LDA can effectively realize the comprehensive analysis of multiple “fingerprint” information. PCA analysis is an internal exploration of data, with only a natural grouping of a multivariate data set and no predictive function. PLS-DA is a supervised discrimination method and is different from PCA. It can integrate the basic functions of multiple linear regression analysis, canonical correlation analysis and principal component analysis and can solve the multicollinearity problem in linear regression (Brereton & Lloyd, 2014). PLS-DA assigns groups to the samples before classification. After grouping, the model adds an implicit virtual binary matrix as the response sample category (specifying a group as 1, and all other values are 0). The multivariate data set is the independent variable matrix used to train the model, the prediction sample independent variable data set is assigned through the threshold of the training model, and the sample category is defined. After establishing the A. sinensis PLS-DA model, the classification performance of the model and the model itself were evaluated. R2 is the regression coefficient used in the PLS-DA model, which was used to evaluate the overall fit of the model. The recommended value of R2 for a good model is between 0.6–1. Q2 refers to the predictive ability of the model after modeling, and should be between 0.5 and 1 (Ballabio & Consonni, 2013; Albergamo et al., 2018). The Variable Importance Plot (VIP) was used to measure the variables with high contribution of the projection on the first two axes (Sayago, González-Domínguez & Beltrán, 2018). LDA is also a dimensionality reduction technique of supervised learning. LDA reduces the dimensionality of samples by maximizing the difference between groups and minimizing the difference within the group. It depends on the average within the group during dimensionality reduction. LDA and PLS-DA have the same data verification method. First, the model was verified by modeling its own data set. Due to the limited survey data, we used the leave-one-out method (LOOCV) to verify the predictive ability of the built model. Finally, statistics of real samples and model prediction samples were represented by a confusion matrix, and the model prediction accuracy rate was calculated. PCA, PLS-DA and LDA are traditional statistical discriminant methods, which usually perform well in terms of geographic origin classification when excluding redundant/confounding predictive factors (Gonzalvez, Armenta & De La Guardia, 2009). They have been widely used in the discriminant analysis of food and drug origin traceability.

The significant differences between each variable in the Linxia, Gannan and Dingxi samples and multiple comparisons of Tukey-HSD were determined using the psych package in R software (version 3.6.1). PCA was conducted using the FactoMineR and factoextra packages in R; PLS-DA model modeling, evaluation, and verification were determined by SIMCA software (version 13.0; Umetrics AB, Umeå, Sweden). The modeling and evaluation of the LDA model were conducted using SPSS (version 21.0; Chicago, USA).

Results

Multi-element and stable isotope characteristics of A. sinensis

Statistical analysis of the eight mineral elements and three stable isotope ratios of A. sinensis sampled from three regions were determined using ICP-MS and IRMS. Results are presented in Table 1. Among them, the K content (5639.95 ± 311.94 mg/kg) in 25 A. sinensis samples was the highest, and the variation range and variance were the largest, followed by Ca (869.37 ± 34.70 mg/kg), Mg (751.88 ± 34.34 mg/kg), Al (119.13 ± 10.60 mg/kg), Zn (5.32 ± 0.22 mg/kg), Mn (5.27 ± 0.22 mg/kg), Cu (1.74 ± 0.08 mg/kg) and Cr (1.02 ± 0.11 mg/kg). K, Ca, Mg, Zn, Mn and Cu were moderately variable, while Al and Cr were highly variable. The stable isotopes δ13C, δ15N and δ18O were - 23.79 ± 0.1‰, 2.13 ± 0.39 ‰and 25.07 ± 0.22 ‰, respectively. The highest coefficient of variation of δ15N is 0.91, and the coefficient of variation of δ13C and δ18O were lower than 0.1. The variation of Al, Cr and δ15N were highly variable. Therefore, the greater the difference between elements among A. sinensis, the more likely it will be suitable for origin traceability.

Table 1 Overview of the mineral elements and stable isotopes of A. sinensis from Linxia, Gannan and Dingxi.

	Range	Median	Mean	SE	std.dev	coef.var	
K(mg/kg)	2806.75–8234.19	5958.66	5639.95	311.94	1559.71	0.28	
Mg(mg/kg)	423.82–1191.03	748.98	751.88	34.34	171.69	0.23	
Zn(mg/kg)	3.41–7.70	5.06	5.32	0.22	1.12	0.21	
Cu(mg/kg)	1.02–2.55	1.67	1.74	0.08	0.39	0.22	
Ca(mg/kg)	511.78–1333.02	889.09	869.37	34.7	173.52	0.2	
Mn(mg/kg)	3.50–7.86	5.18	5.27	0.22	1.12	0.21	
Cr(mg/kg)	0.50–2.73	0.9	1.02	0.11	0.57	0.56	
Al(mg/kg)	66.38–287.77	106.49	119.13	10.6	52.99	0.44	
δ13C(‰)	−24.83–−22.75	−23.94	−23.79	0.1	0.52	−0.02	
δ15N(‰)	−2.08–5.57	1.91	2.13	0.39	1.95	0.91	
δ18O(‰)	23.44–27.10	24.77	25.07	0.22	1.08	0.04	
Notes.

Range variable range

median median values from the three regions

mean average values from the three regions

SE standard error

std.dev standard deviation

coef. Var coefficient of variation

The coefficient of variation (CV) measures the degree of variation within an element (less than 0.2, has a low degree of variation; between 0.2 and 0.3, a medium variation; greater than 0.35, highly variable).

The mean ± standard error of Linxia, Gannan and Dingxi in Gansu Province were calculated to compare the differences between elements from the three regions (Table 2). Compared with the mineral elements in the three regions, Linxia A. sinensis K was significantly lower than Gannan and Dingxi (P < 0.05), while Al and Ca/Al were significantly higher than Gannan and Dingxi (P < 0.05). Mg, Zn, Cu, Ca, Mn, Zn/Mn and Cu/Cr had no significant difference. Among the stable isotopes, the δ13C and δ15N from Linxia were significantly (P < 0.05) lower than that of Gannan and Dingxi, and the δ18O from Gannan was significantly (P < 0.05) higher than that of Linxia and Dingxi.

Table 2 Statistical analysis of the mineral elements and stable isotopes (mean± SE) of A. sinensis sampled from Linxia, Gannan and Dingxi.

	LX	GN	DX	
K(mg/kg)	3562.27 ± 305.41a	6396.38 ± 298.33b	6031.74 ± 401.89b	
Mg(mg/kg)	674.73 ± 80.74a	689.48 ± 32.96a	815.15 ± 52.04a	
Zn(mg/kg)	5.97 ± 0.76a	4.84 ± 0.20a	5.32 ± 0.29a	
Cu(mg/kg)	2.05 ± 0.16a	1.54 ± 0.14a	1.72 ± 0.09a	
Ca(mg/kg)	769.65 ± 86.01a	827.00 ± 30.92a	930.54 ± 52.11a	
Mn(mg/kg)	6.13 ± 0.54a	5.36 ± 0.39a	4.90 ± 0.28a	
Cr(mg/kg)	1.59 ± 0.32b	0.74 ± 0.11a	0.95 ± 0.14ab	
Al(mg/kg)	186.52 ± 36.04b	97.98 ± 11.73a	104.61 ± 6.72a	
δ13C(‰)	−24.39 ± 0.15a	−23.6 ± 0.15b	−23.66 ± 0.13b	
δ15N(‰)	−0.15 ± 0.66a	3.71 ± 0.52b	2.17 ± 0.42b	
δ18O(‰)	24.65 ± 0.30a	26.38 ± 0.16b	24.52 ± 0.23a	
Notes.

Significant differences were analyzed using the ANVOA-Tukey HSD method.

Different letters indicate significant differences in the three regional variables (P < 0.05).

LX Linxia

GN Gannan

DX Dingxi

PCA of A. sinensis

The original data (K, Mg, Zn/Mn, Cu/Cr, Ca/Al, δ13C, δ15N, δ18O) were normalized and analyzed by factor analysis. The Kaiser-Meyer-Olkin (KMO) test and Bartlett sphere test results were KMO = 0.506 (KMO > 0.5) and P = 0.045 (P < 0.05). The approximate chi-square value of Bartlett’s sphere test was 41.83, which satisfies sampling adequacy and Bartlett’s sphere test significance. Table 3 shows that in PCA analysis, the first principal component explained 38.42% of the total variance, the second principal component explained 20.31% of the total variance, and the first three principal components explained 73.68% of the total variance. These results can fully reflect the main data.

Table 3 Principal component analysis of the first three-axis eigenvalues and variance interpretation rate.

	Eigenvalue	Variance percent (%)	Cumulative variance percent (%)	
Dim.1	2.95	38.42	38.42	
Dim.2	1.56	20.31	58.73	
Dim.3	1.15	14.95	73.68	

The relationship between the principal component and each element was explored through the factor loading plot and the contribution rate of each element on the first five principal component axes. We examined the significant correlation (P < 0.05) between principal component axes and mineral elements and stable isotopes (Fig. 2). Figures 2A and 2B show that Ca/Al, K, δ13C, and δ15N had a significant positive correlation (P < 0.05) with the first principal component axis. Zn/Mn and Cu/Cr had a significant positive correlation (P < 0.05) with the second principal component axis, and δ18O had a significant negative correlation (P < 0.05) with the second principal component axis. The contribution of mineral elements and stable isotopes on the first five principal component axis was explained. Ca/Al, K, δ13C and δ15N had a high contribution to the first principal component axis, while Zn/Mn, Cu/Cr and δ18O had a high contribution to the second principal component axis. In addition to the first two axes, the other axes have a low degree of interpretation of population variance, and K, Mg, δ13C, δ15N, δ18O, Zn/Mn and Cu/Cr have strong correlations in other principal component axes. Therefore, the explanations of contributions to other axes cannot be trusted. The histogram in the lower right corner of Figs. 2C and 2D quantifies the contribution of mineral elements and stable isotopes in the first two axes, and provides a threshold for the contribution of higher variables.

Figure 2 The relationship between mineral elements, stable isotopes and principal components in PCA.

(A) Factor loading plot; (B) correlation plot between the principal components and each factor; (C) mineral elements and stable isotopes contributions on the first principal component axis; (D) mineral elements and stable isotopes contributions on the second principal component axis.

The distribution of principal component scores of A. sinensis based on K, Mg, Zn/Mn, Cu/Cr, Ca/Al, δ13C, δ15N, and δ18O in the three regions on the first two principal component axes are shown in Fig. 3A. The score diagram shows that the first principal component axis can distinguish A. sinensis sampled from Linxia and other places. Since Ca/Al, K, δ13C and δ15N contributed more to the first principal component axis, Ca/Al, K, δ13C and δ15N are important factors to distinguish A. sinensis from Linxia and other places. It also suggests that the Ca/Al, K, δ13C, and δ15N in Linxia A. sinensis are lower than that of Dingxi. Although there is no clear distinction between Dingxi and Gannan, most of the samples fall into the first and fourth quadrants, respectively, suggesting that A. sinensis from Dingxi had higher values of Zn/Mn and Cu/Cr, while A. sinensis from Gannan had a higher δ18O value.

Figure 3 (A) A. sinensis sample scores from Linxia (LX), Gannan (GN) and Dingxi (DX) on the first two principal component axes; (B) A. sinensis sample scores from Gannan (GN) and Dingxi (DX) on the first two principal components.

A. sinensis from Dingxi and Gannan are not easily classified (Fig. 3A). This may be due to the close geographical distance between Dingxi and Gannan. Taking into consideration that there are differences in the centroid of A. sinensis in Dingxi and Gannan on the second principal component axis, and that the contribution rates of Zn/Mn, Cu/Cr and δ18O on the second principal component axis are high, three factors of Zn/Mn, Cu/Cr and δ18O were used in the PCA of A. sinensis sampled from Dingxi and Gannan. Figure 3B shows A. sinensis scores from Dingxi and Gannan, and the first two axes explain 90.23% of the total variance. The first principal component axis was mainly affected by Zn/Mn and δ18O, and the second principal component axis was mainly affected by Cu/Cr. The samples of A. sinensis from Dingxi and Gannan can be clearly classified.

PLS-DA of A. sinensis

We analyzed 25 samples of A. sinensis from Linxia, Gannan and Dingxi based on eight factors (K, Mg, Zn/Mn, Cu/Cr, Ca/Al, δ13C, δ15N, δ18O). Samples were analyzed using PLS-DA, and the first three component axes were extracted with eigenvalue ≥1. Table 4 shows that the first three component axes explained 70% of the total explanatory variables (the first axis explained 38%, the second axis explained 17%, and the third axis explained 15%). Fitting the response variables, the first three axes explained 68% of the total response variables (the first axis explained 29%, the second axis explained 36%, and the third axis explained 3%). In this model, the cumulative Q2 of the first two axes was 0.52 (suggested Q2 > 0.5), the negative effect of the third axis on the model’s predictive ability was slightly reduced, and the cumulative Q2 was 0.47.

Table 4 The main parameters of the first three-axis fitting of the PLS-DA model.

Component	R2X	R2X(cum)	Eigenvalue	R2Y	R2Y(cum)	Q2	Q2(cum)	
1	0.38	0.38	3.06	0.29	0.29	0.19	0.19	
2	0.17	0.55	1.33	0.36	0.65	0.41	0.52	
3	0.15	0.70	1.21	0.03	0.68	−0.15	0.47	

We reduced the dimension of the explanatory variables and the scores of these variables are shown in Fig. 4A. t1 and t2 are the first and second component axes of the explanatory variables after dimensionality reduction. These explain 55% of the total variable variation. A. sinensis in the Linxia, Gannan and Dingxi regions are separated and clustered into one category, respectively. We were able to determine the importance of the mineral elements and stable isotopes of A. sinensis in PLS-DA. We also determined the correlation between variables and variables and variables and sampling areas and analyzed the PLS-DA model loading graph (Fig. 4B). K, Cu/Cr, Ca/Al, δ13C, and δ15N had larger loading values in the first component, and had a negative correlation. The samples of A. sinensis from Linxia were distant from the sampling areas, so these five variables were in Linxia and the samples of A. sinensis from Gannan and Dingxi differed. Mg, Zn/Mn, and δ18O had higher loading values in the second component, and the second axis was positively correlated with Mg and Zn/Mn, and negatively correlated with δ18O. Since the A. sinensis samples from Dingxi and Gannan were far apart on the second axis, these three variables are important variables for separating Dingxi and Gannan.

Figure 4 (A) PLS-DA model’s scatter score graph in different regions; (B) factor loading graph of the PLS-DA model in different regions (LX, Linxia; GN, Gannan; DX, Dingxi).

In the factor load diagram, w*c [1]/[2] is used to measure the mineral elements and stable isotopes load, where w* is the weight of the standardized explanatory variable matrix obtained from the component t, and c is the weight from the component u (the weight of the response variable in the variable dimension reduction matrix). Explanatory variables near the response variable are able to distinguish data between plots.

We analyzed the variable component axis of the projection in the PLS-DA model, and filtered out the important variables of the projection of the entire model (Fig. 5A). In the A. sinensis samples, the VIPs of the five variables (δ18O, Zn/Mn, δ15N, K and Ca/Al) were all greater than 0.9 (recommended is VIP > 0.5). This indicates that these are important variables for classification in the PLS-DA model and are of great significance to the model interpretation and source traceability of A. sinensis.

Figure 5 (A) The variables importance for the projection (VIP) for the A. sinensis PLS-DA model in different regions; (B) RVY2 and QVY2 values in the PLS-DA cross-validation model.

In order to evaluate the fit and prediction performance of the cross-validation set of the built A. sinensis PLS-DA model, R2VY was used to evaluate the fit of the cross-validation model, and Q2VY was used to evaluate the predictive ability of the cross-validation model. In order to prevent the samples from overfitting in the PLS-DA model, 200 permutation tests were performed on the first three component axes of samples from Linxia, Gannan and Dingxi. These tests evaluated the validity of the cross-validation model and the stability of the model fitting. Our results show that the R2VY of Linxia, Gannan and Dingxi were 0.62, 0.76, and 0.65, respectively, and the Q2 VY were 0.41, 0.59, and 0.39, respectively, which reflects the good fit and predictive ability of the cross-validation training set to the PLS-DA model (Fig. 5B). We applied 200 permutation tests on the first three axes of samples taken from Linxia, Gannan and Dingxi, which indicated that the built PLS-DA cross-validation model showed strong validity.

Finally, internal verification was carried out on A. sinensis samples from Linxia, Gannan and Dingxi based on the PLS-DA model. Due to the small data set, the original data verification and leave-one-out cross validation were used to verify the A. sinensis samples (Table 5). The original data successfully verified the samples of A. sinensis from Linxia, Gannan and Dingxi, and the accuracy rate of all the samples was 100%. We used leave-one-out cross validation to verify the samples, and the total corrected rate of all of the A. sinensis samples was 84%. The verification rate of A. sinensis from Linxia was only 60%, indicating that two samples of Linxia A. sinensis were wrongly classified as being from Dingxi. The verification rate of A. sinensis samples from Gannan was 100%, and all classifications were correct. The correct rate of the A. sinensis discriminant from Dingxi was 84.62%, of which 2 A. sinensis samples were wrongly judged as being from Linxia. This may be related to the geographical scale of Dingxi, and the large variation of variables among the species.

Table 5 A. sinensis classification results from the PLS-DA model.

Procedure	Provenance	Expected belonging groups	Correct	
		LX	GN	DX		
Original	Linxia	5	0	0	100%	
	Gannan	0	7	0	100%	
	Dingxi	0	0	13	100%	
	Total				100%	
Leave-One-Out Cross Validation	Linxia	3	0	2	60%	
	Gannan	0	7	0	100%	
	Dingxi	2	0	11	84.62%	
	Total				84%	
Notes.

LX Linxia

GN Gannan

DX Dingxi

LDA of A. sinensis

Stable isotopes and mineral elements were analyzed using LDA. Our analysis found that the three A. sinensis samples were clustered together. We also found certain differences between the groups. The recognition ability of the three stable isotopes for the two discriminant functions (LD1, LD2) explained 100% of the total variance (discrimination function 1 explained 71.8% of the total variance, and discriminant function 2 explained 28.2% of the total variance; Fig. 6). The function test was performed after the LDA model was established. The Wilks’ lambda value of the two discriminant functions was 0.073, and the difference between the groups was significant (P < 0.001). The discriminant functions of Linxia, Gannan and Dingxi were as follows:

Figure 6 LDA model for A. sinensis samples from different regions (LX, Linxia; GN, Gannan; DX, Dingxi).

Y(LX) =0.014K −0.069Mg−174.241 δ13C −19.921

δ15N+55.035 δ18O+16.287Zn/Mn+11.272Cu/Cr+5.545Ca/Al−2836.829

Y(GN) =0.014K−0.067Mg−167.540 δ13C−

15.833 δ15N+59.811 δ18O+11.835Zn/Mn+16.983Cu/Cr+4.480Ca/Al−2802.584

Y(DXMX) =0.014K−0.059Mg-169.654 δ13C-

18.146 δ15N+54.145 δ18O+18.390Zn/Mn+13.029Cu/Cr+5.614Ca/Al−2720.159

We used the original data set and the leave-one-out test set to verify the built LDA model. The accuracy of the LDA model in the original data sets of Linxia, Gannan and Dingxi was 92% (Table 6). Among them, one sample of Linxia was wrongly identified as being from Dingxi, and two sample from Dingxi were wrongly identified as being from Gannan. The correct discrimination rate was 80% using leave-one-out cross validation. The misjudgment was the same as the original data set self-validation.

Table 6 A. sinensis classification results from the LDA model.

Procedure	Provenance	Expected belonging groups	Correct	
		LX	GN	DX		
Original	Linxia	4	0	1	80%	
	Gannan	0	7	0	100%	
	Dingxi	1	0	12	92.31%	
	Total				92%	
Leave-One-Out Cross Validation	Linxia	3	0	2	60%	
	Gannan	0	7	0	100%	
	Dingxi	2	1	10	76.92%	
	Total				80%	
Notes.

LX Linxia

GN Gannan

DX Dingxi

Discussion

A. sinensis was discovered in Gansu and is recognized as high-quality medicinal material. In 2017, A. sinensis was listed as a protected product in Minxian County, Gansu. Where specimens originate plays an important role in the formation of authentic medicinal materials. Blindly introducing authentic medicinal materials may cause a decline in quality and result in fake or inferior medicinal materials used in the production and sale of medicinal products. This would in turn reduce the safety and efficacy of Chinese medicinal materials. Food and drug quality and safety issues are closely related to health and consumers rights and involve multiple interests in the entire process. Recently, frequently occuring food and drug safety incidents has aroused public attention, which affects mutual trust in trade and threatens the safety of clinical medication. It is important to address the best ways to control and assess the safety of food and drugs. Food and drug safety be monitored at every point in the food supply chain from planting and harvesting to receiving. Meanwhile, the long-term development of geographical products requires strict management, and the application of scientific and technical means for monitoring and inspection. These methods will ensure the protection of producer and consumers interests and a sustainable development of the industry. A. sinensis from Gansu Province requires an effective tracing method. The traceability of the authentic medicinal material will ensure control over their safety, efficacy and provenance.

Mineral elements and stable isotopes, as geographical indicators in plants, are closely related to factors such as climate, mineralogy, element mobility, bioavailability and physiological adaptability of the species. The mineral elements and stable isotopes in plants vary in different environments as determined by the plant’s adaptation to the environment and the interaction between the environment and plants. The fingerprints of mineral elements and stable isotopes in plants are the imprint of the environment. Wang et al. (2020) traced the origin of 25 mineral elements in maize from three provinces and determined that the accuracy of eight elements using the SLDA model was 92.2%. Pereira et al. (2018) traced the origin of Arapaima spp. fish from four different regions using strontium and carbon isotopes. Their results show that the accuracy rate of identification of wild and aquaculture fish was 58% and that of four different regions was 76%. We collected mineral elements and stable isotopes to trace the geographical origin of A. sinensis and determine differences between the geological and climatic environments in the three A. sinensis-producing areas. Based on the significant differences and the results of our model, K, Ca/Al, δ13C, δ18O and δ15N were determined to be important factors to distinguish A. sinensis in these regions.

We compared the discriminant analysis of A. sinensis using PLS-DA and LDA. The original data set was used to verify the model, and we found that the accuracy of the PLS-DA model was higher than that of the LDA model. The model was verified by the leave-one-out, and the accuracy rate of the PLS-DA model was higher than that of LDA model. The self-verification of the original data set to the model was a verification of the built model itself. The original data sets involved in the modeling process and the information from the verified original data set were included in the verification as the model was built. Leave-one-out cross validation is suitable for small sample analysis. A single prediction sample was removed during model modeling, the model was trained on all remaining samples, and then all samples were predicted. The total variance explained by the first two axes of the LDA model was higher than that of the PLS-DA model. Model verification using the leave-one-out method showed that PLS-DA was more correct than LDA. The PCA model is suitable for exploring the natural classification of variables in A. sinensis from Linxia, Gannan and Dingxi. The PLS-DA model evaluated the importance of variables and classified the samples. The PLS-DA model was also better than the LDA model in terms of classification and discrimination. Our study integrated mineral elements and stable isotopes to trace the origin of A. sinensis from three regions of Gansu Province. We used the PCA model to explore the data naturally, and the PLS-DA model and the LDA model to leave-one-out cross-validation. The comprehensive accuracy rates were 84% and 80%, respectively.

Conclusion

Differences between mineral elements and stable isotopes of A. sinensis sampled from Linxia, Gannan and Dingxi, and Gansu Provinces were found and compared. Significant differences were found in K, Cr, Ca/Al, δ13C, δ15N, and δ18O (P < 0.05).

Significant differences were found among the three groups. PCA and PLS-DA models showed that K, Zn/Mn, Ca/Al, δ13C, δ15N and δ18O were important variables for distinguishing the three regions. However, only K, Ca/Al, δ13C, δ18O, and δ15N demonstrate significant differences among the three regions (P < 0.05). K, Ca/Al, δ13C, δ18O, and δ15N play an important role in the discriminant analysis of A. sinensis among the three regions.

The results of mineral elements and stable isotopes from the PLS-DA and LDA show that the discriminant accuracy rate of the PLS-DA model was 84% and the accuracy rate of the LDA model was 80%.

Supplemental Information

Supplemental Information 1 Sampling coordinates

Click here for additional data file.

Supplemental Information 2 The results of permutation test of A.sinensis in the first three axes of PLS-DA model in Dingxi area

Click here for additional data file.

Supplemental Information 3 The results of permutation test of A.sinensis in the first three axes of PLS-DA model in Gannan area

Click here for additional data file.

Supplemental Information 4 The results of permutation test of A.sinensis in the first three axes of PLS-DA model in Linxia area

Click here for additional data file.

Supplemental Information 5 Raw Data

Click here for additional data file.

We would like to thank those involved in this project and the reviewers who provided constructive comments. We would also like to thank Ling Jin, who provided experimental materials.

Additional Information and Declarations

Competing Interests

Author Contributions

Field Study Permissions

Data Availability

The authors declare there are no competing interests.

Shanjia Li conceived and designed the experiments, analyzed the data, prepared figures and/or tables, authored or reviewed drafts of the paper, and approved the final draft.

Hui Wang conceived and designed the experiments, performed the experiments, analyzed the data, prepared figures and/or tables, authored or reviewed drafts of the paper, and approved the final draft.

Ling Jin, James F. White and Kathryn L. Kingsley conceived and designed the experiments, authored or reviewed drafts of the paper, and approved the final draft.

Wei Gou, Lijuan Cui, Fuxiang Wang and Zihao Wang performed the experiments, prepared figures and/or tables, and approved the final draft.

Guoqiang Wu analyzed the data, authored or reviewed drafts of the paper, and approved the final draft.

The following information was supplied relating to field study approvals (i.e., approving body and any reference numbers):

Our sampling work was completed during the Angelica harvest season, and the harvest location was in the farmers’ fields. They are Xingzhen Wang, Xiaoxia Liu, Fengqiang Zhang, Baoguo Shi, and Shufen Wang. Researchers could directly contact farmers and obtain their permission to collect experimental samples.

The following information was supplied regarding data availability:

The data are available in the Supplemental Files.

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
