# Peer review of "Validation and analysis of the geographical origin of Angelica sinensis (Oliv.) Diels using multi-element and stable isotopes"

_PeerJ, doi:10.7717/peerj.11928_

## Round 0.1 · original submission · Major Revisions

Both reviewers provide a number of detailed comments and suggestions that will be useful in revising and improving the manuscript. I agree with both reviewers that much of the text in the Discussion section is not appropriate for that section. I also agree with reviewer 1 that the Results section is much too long relative to the amount of data that were collected and reported. The authors also must clarify their stable isotope methods, as noted by reviewer 1, particularly in regards to the standards that were used and the importance of using at least two standards per isotope for data normalization. However, I leave it up to the authors to decide whether to add S or H isotope data.

Reviewer 1 ·

Basic reporting

Overall, the writing of this submission needed at least another round or two of edits to meet the minimum standards of PeerJ. Throughout the manuscript there are many grammatical and spelling errors that effect the clarity of the study. Unfortunately, these are too numerous to list here but some examples:

Line 18: Background section of abstract is very confusing.
Line 58: What is the difference between “human material exchanges” and the import and export trade? Aren’t this the same thing?
Line 83: “The fingerprint of mineral elements and stable isotopes in plants is a unique "brand" in environments.”
Line 106: “semi-humid-mid-temperate semi-arid” too many hyphens
Line 358: “most studies have shown that the mineral elements in the soil are significantly related to their mineral elements “
Lines 369 and 372: “the relative size of CO2 in the external environment and the CO2 concentration between plant cells” and “The size of stable oxygen isotopes in plants”. I believe the author is referring to the δ13C and δ18O values?

Many of the grammatical issues could be fixed with some minor revisions, however, the structure of the paper deeply flawed. The manuscript’s introduction doesn’t do a good job establishing relevant background for the study nor does it provide the reader with reason to be invested in the research. It is not clear to me why the provenance of A.sinensis in the Gansu Province is important. The first two paragraphs of the Introduction try to establish this importance but provide only tangential information. While it is important to put A.sinensis in a cultural and economic context, these two paragraphs could probably have been cut to ¼ in length. In fact, the discussion does a significantly better job at establishing relevant background for this study. For example:

Line 319-351: These paragraphs read more as “Introduction” and should be moved to that section. Much of this section is establishing other studies of “food provenance”. However, I think some of these examples that do not involve stable isotopes or mineral elements could be viewed as tangential and could be excised.

Line 356-385: Much of these sections could be condensed and moved to the Introduction
Additionally, a section of the discussion (Lines 390-423) could be condensed and moved to the methods under “Statistical Analysis” as it justifies the statistical tools used in this manuscript. Unfortunately this leaves very little for the discussion itself which would then need to be further fleshed out.

The statistical results take up a significant part of the manuscript. This likely could be more concise.

The figures and tables have several formatting problems:

Figure 1: The legend shouldn’t be intersected by the Latitude and Longitude. The font of all the text is too small to read. The font should be 12pt. The scales for provincial and regional maps are too small. The provincial map shouldn’t have Latitude and Longitude lines or its own tiny legend. It is also too small to have the regions of the Gansu province named in the smaller map. Names of regions should be indicated in all the areas of the larger map. Large arrow should not be necessary with adequate Figure description.

Figure 2: The font of all the text is too small to read. The font should be 12pt. All sections of the figure could be put into their own boxes to make it a bit clearer. Some of the text seems scrunched like the text at the far right of 2B.

Figure 3: The font of all the text is too small to read. The Legend is too small. The individual datapoints are difficult to differentiate and should be larger. The “group centroid” datapoint should also be included in the legend.

Figure 4: The font of all the text is too small to read. The Legend is too small. Axes could be thicker to increase clarity. Subfigure letters should probably be of a consistent size across figures.

Figure 5: The font of all the text is too small to read. The Legend is too small. Axes lines and error lines could be thicker. The shading on the bars should be removed. It appears that this figure has been compressed vertically.

Figure 6: The font of all the text is too small to read. The whole figure appears fuzzy. It is not at the proper DPI. Sample icons should be the same across all figures. The colors are the same for each region across figures but Figures 1 and 3 use different shapes while Figures 4 and 6 only use circules.

Table 1: The font of all the text is too small to read. “n” is unnecessary in the table given that all variables are n = 25. Min/max and range should be combined into one line (e.g., for K: 2806.75 – 8234.19). Multiple metrics of variation included here seem unnecessary. This table can be pared down to only essential elements for the results and discussion.

Table 2: No description of what “a” and “b” superscript mean in the description of table. Type of statistical analyses should be Table description and not a note.

Table 3: Lots of empty space. Could this be redesigned to be smaller.

Table 5: Provenance is spelled wrong. There isn’t a need to capitalize the letters in the middle of the region names.

Table6: Same as Table 5

Experimental design

The primary research within this manuscript does fall within the aims and scope of PeerJ. The authors are relatively clear that this study provide geochemical tools to determine provenance of A.sinensis. However, they do not fully establish why this information is important.

I’ve touched up on “Statistical analysis” subsection of the methods previously and the “Study area and sample collection” provides adequate information. The “mineral elements analysis” subsection also appears to provide a good concise description. However, there are major concerns with the “Stable isotope measurement” subsection:

Line 119-122: I believe the authors are trying to say they are using the δ13CVPDB, δ15NAir and δ18OV-SMOW scales for their stable isotope analyses. The way that it is written implies they used VPDB, Air and SMOW as standards within their stable isotope analyses.

Line 123: “wheat flour δ13C (δ13CVPDB=-13.68±0.2 (‰)) and δ15N (δ15NAir=1.58±0.15 (‰))” These are the same values stated earlier for the international scale. We are not given the value of their internal wheat standard nor its origin. Is it a standard verified by USGS or IAEA? An internal standard calibrated with verified stable isotope standards?

Line 124: The authors are only using a single standard for their stable isotope analyses. It is recommended to use at the minimum two stable isotope standards with values representing the range of expected values within analyses. I would be wary to trust this data completely. Furthermore, there likely should have been a standard weighed at different masses to correct for amplitude and one or more standards for independent checks. The authors also do not indicate if there were any replicates in the analyses.

Line 126: The author does not provide what standard or standards they used for δ18O analyses.

Line 128: The author does not provide the target weights for their carbon and nitrogen stable isotope analyses.

Line 131-132: It is not mentioned that samples for δ18O analyses should be placed silver cups. The authors also do not provide the peripheral that was used for the analyses, only the IRMS. TC/EA? FlashEA?

Line 132-138: Description on how delta values are calculated is unnecessary.
I am also curious as to why the authors did not generate data for δ34S or δ2H. Both are used in other food provenance studies (e.g., Camin et al 2017). While sulfur has its complications and the author’s FlashEA might not be set up for sulfur, δ2H is typically measured simultaneous with δ18O. The additional isotopes could have provided a much more robust analysis.

Validity of the findings

While modelling is not field of expertise, the statistical analyses and the results presented in this manuscript appear relatively sound and follow what has been established in earlier provenance studies. However, I would like to reiterate that the results section is hard to follow and need significant streamlining to brevity and clarity.

The authors did include their data but it lacks GPS coordinates for each sample and having the values for the standards would provide us more confidence in the data.

Any misgivings I’d have for the validity of the findings are related in the lack of robustness in the stable isotope analyses. From the methods it is hard for me to determine if the authors have a firm grasp of the process of stable isotope analysis. This is not helped by the lack of information and the misinformation in that section of the methods. I would have preferred to see additional use of isotopic standards from carbon and nitrogen along with perhaps the inclusion of hydrogen and sulfur if feasible. This being said, the authors were able to effectively model the provenance of A.sinensis using just 25 individuals across the 3 regions.

The final conclusions for the manuscript are succinct and effectively summarize the results but don’t tie in very well to introduction or the broader impact. Again, from the manuscript, it is difficult to know why the provenance of A.sinensis is important economically, culturally or medically.

·

Basic reporting

The manuscript is written in clear language, having a logical path for information from introduction, methods and results collection. Nonetheless, for the first part of the discussion paragraph, it is advisable to be moved under introduction as they refer to other studies on geographical traceability which usually it should be under the introductory or bibliography part of the manuscript.
Under “discussion” it should be limited to make comparison of collected data from previous studies

Such study is mainly based on the application of statistical analysis as tool to extract evidence for traceability, thus it is highly recommended to explain the objective of each tool used to extract data in the paragraph of “Statistical Analysis”, as the use of some of those tools was not very clear in interpretation of relationship between different variables, besides that many tools were used along the whole study



The listed references are relevant to the topic and considered between state of the art references for geographical traceability; besides that most references dated from less than 10 years

All presented figures, tables together with their labeling are very well clear and most data are reflected in the discussion part.

Experimental design

The study was designed in structured way; collecting samples from 3 regions as origin of the plant Angelica Sinensis. Knowing that number of samples should be higher as 7 samples from one area could not be enough to reflect the representativity of the soil type which has an impact on tracing the origin of samples.

Validity of the findings

data reported in this study matches previous studies in term of statistical analysis tool and interpretation,

Regarding analytical methods, the author used traditional methods applicable to such type of study based on stable isotopes and total concentration of selected elements.

The missing part is quality control data for total element analysis, as it should be part from evaluation of credibility of data

Additional comments

Few notes are to mention and to be considered by author

- Page 8 line 120: please check the values of CRM Vienna pee Dee belemnite (δ13CVPDB=-13.68‰),Air (δ15NAir=1.58‰) and Vienna
- line 148: usually during acid digestion we need to take good care not to loose any volatile elements… so lid should stay closed till the sample get to the ambient temperature
- line 159; the author mentioned “A small number of LOQ…” ; it is not clear why they decided to devide by square root of 2? What does it mean “A small number…” how many ?? why not all ?? it needs to elaborate more on this sentence
- LoQ for stable isotopes? Can the author elaborate how to compute this parameters for stable isotopes???
- Line 179: do the stable isotopes values reported are the average of all samples from each region??
- Line 180: the values of coefficient of variation is related to the variation within the samples from each region or from the 3 regions??
- Line 178 the author mentioned already that variation of Al and Cr are high; again in line 181 same idea is repeated but including δ15N; so it is better to rearrange the results 1- per values level, 2- per variability within the group
- What the author meant by presenting the ration between only selected elements Zn/Mn, Cu/Cr… while not for others as it was not reflected in the text and the objectives from this ratio
- The variation δ13C of is usually not related to the level of precipitation while it is applicable for O18 and Deuterium
- Line 360-363 it needs rewording , very complicated sentence
- Table 2 labelling: what the author meant “with statistical analysis data … although the values reported are the concentration of total elemnts and Delta values for N, O and C

---

## Round 0.2 · Minor Revisions

One of the prior reviewers has re-reviewed this manuscript and has found it much improved. However, the reviewer also noted that the manuscript still contains many grammatical and spelling errors. The reviewer notes some examples for improvement, but the list from the reviewer is not exhaustive. For example, in the revised abstract I found at least 4 simple mistakes (listed below) that must be fixed. There are similar issues throughout the manuscript. Please note that "clear, unambiguous, technically and grammatically correct English" is a requirement for publication in PeerJ.

Line 17: Authentic does not need to be capitalized.

Line 19: Industrial is misspelled.

Lines 22-28: This is a run-on sentence. Please break it up into at least 2 sentences.

Line 34: There is a space missing after "regions".

Reviewer 1 ·

Basic reporting

The manuscript as a whole is greatly improved from the first draft submitted. There are still a number of issues with spelling and grammar but these are minor compared to the structural issues in the original submission. I've provided some examples below but these are far from exhaustive. There needs to be another serious pass here on grammar and spelling.

Line 53: Maybe say “effective scientific tools and techniques”?
Line 61: provenance is a good term in place of “source authenticity”
Line 61-63: I understand what you are trying to say here but it might not be clear to all readers. I assume you mean that the phylogeography can provide interrelatedness among natural populations but not necessarily the geographic origin of cultivated plants?
Line 64: remove trace of
Line 78: components instead of ingredients
Line 79: tool instead of target
Line 80: “Stable isotopes of plants” instead of stable plant isotopes
Line 82: d13C values instead of “stable carbon isotope”. You can also use “stable
carbon isotope values”
Line 83-84: A bit hard to follow. Do you mean the pCO2 between the external environment and within the plant leaf/mesophyll? I understand what you are trying to say again but it could use more clarity.
Line 85 you can probably drop the sentence “When the interceullar…”
Line 88-89: You can mention isoscapes also?

Line 123: awkward sentence. “We weighed 5.0 mg of homogenized/ground A sinensis into tin capsules” or something similar would read better.

Line 127: “To bring our samples to international stable isotope values…etc” instead of “Use USGS24…”

Line 129 and 139: You can probably drop the sentences with VSMOW and VPDB

Line 130: “long term accuracy/precision” of instrument instead of long term standard deviation for d13C

Line 131: B2159 is a Sorghum Flour std from Elemental Microanalysis I believe. It’s d15N value here is reported correctly but d13C should be -13.78 based on the most recent certificate. (However I understand that you may have a different batch than what is presently available for purchase. You may want to report the batch number and that the standard is from Elemental Microanalysis.

Line 133: The internal B2159 standard doesn’t prevent drift but allows you to drift correct your sample runs.


Line 378-383: This paragraph would fit better in the intro as a justification of using trace elements and stable isotope data.

Lines 384-403: Paragraph need to be tightened up and better tied into the study as a whole.

Figure 1: Move the legend so nothing overlaps with the letters

Figure 2: ABCD are too large. Reduce font size and ensure they don't overlap with the lines between panels.

Figure 3: A and B could be smaller fonts

Experimental design

Overall the authors adequately responded to my critiques.Some structural and accuracy issues are touched upon in "Basic reporting".

Validity of the findings

The authors overall were able to incorporate my critiques into the manuscript in an acceptable fashion.

---

## Round 0.3 · Minor Revisions

Thank you for your thorough revisions to the manuscript. However, there
remain significant typographical errors that prevent me from accepting the manuscript at present. For example, 'Turkey's honey significant differences' should be 'Tukey's honestly significant difference' I believe. Additionally, 'Turkey' instead of 'Tukey' occurs multiple times. Please fix these errors and also give the manuscript another thorough read for any additional grammatical and spelling errors.

---

## Round 0.4 · accepted · Accept

Thanks for the revisions you have made to the manuscript.